# SLIDING WINDOW–BASED Q-ENSEMBLE FOR OFFLINE REINFORCEMENT LEARNING

## ABSTRACT

Offline reinforcement learning aims to learn optimal policies from static datasets, which brings the key challenge of accurately estimating values for out-of-distribution actions. Ensemble-based methods address this issue by aggregating multiple Q-networks to reduce the uncertainty in Q-value estimates. However, previous related studies suffer from the inevitably high correlation among Q-functions, driven by identical architectures, shared inputs, and synchronized Bellman targets. Such correlation reduces the robustness of Q-ensembles, ultimately leading to degraded policy performance. In this paper, we propose sliding window delayed gradient (SWDG), a novel ensemble-based offline RL algorithm that leverages the temporal asynchrony induced by the sliding-window mechanism to dynamically maintain diversity among Q-functions. Meanwhile, to further reduce extrapolation error and correlation, SWDG uses Q-networks tied to the sliding window as delayed-gradient target to compute the temporal-difference (TD) error. We theoretically show that the sliding window mechanism tightens the pessimistic lower bound and enhances temporal decorrelation among Q-functions, while the use of delayed-gradient target further strengthens this guarantee. Our experiments on the D4RL benchmark show that SWDG achieves state-of-the-art performance.

## 1 INTRODUCTION

Offline reinforcement learning (RL) seeks to learn optimal decision-making policies from static datasets, without further interactions with the environment. Offline RL can leverage existing data to learn safely and efficiently, making it suitable for some real-world scenarios Maddern et al. (2017); Gürtler et al. (2023) where online exploration is costly or risky. However, learning without interaction presents a critical challenge: the evaluation of out-of-distribution (OOD) actions often results in extrapolation errors and overestimated value functions Fujimoto et al. (2019).

One common approach for offline RL is to impose constraints that encourage learned policies to remain close to the support set of behavior policies in the offline dataset Prudencio et al. (2023). Several methods have been proposed along this line, such as incorporating behavior cloning objectives to regularize policies Fujimoto & Gu (2021); Kumar et al. (2019) or applying value function regularization to enforce pessimism on out-of-distribution (OOD) data Kumar et al. (2020); Lyu et al. (2022). Although these methods help mitigate extrapolation errors, recent work suggests that such constraints can be overly pessimistic Ghasemipour et al. (2022), as they prevent the agent from approaching any OOD data, regardless of whether it is beneficial or not.

Recent methods An et al. (2021); Lee et al. (2022); Zhang et al. (2024) utilize an ensemble of Q-functions to quantify the uncertainty of Q-values estimation to solve the OOD issue. By estimating uncertainty, these approaches can identify OOD data and assign their Q-values with high confidence. EDAC An et al. (2021) proposes a diversity loss to avoid strong correlations between critics, aiming to improve the reliability of uncertainty estimation. MSG Ghasemipour et al. (2022) and EQL Zhang et al. (2024) argue that previous ensemble methods tend to produce pessimistic estimates due to the use of shared pessimistic targets, thus they utilize an ensemble of Q-functions with independent target to obtain an accurate lower confidence bounds (LCB) of Q-values. The theory assumption of these method is that individual Q-functions are mutually independent. However, this assumption is difficult to satisfy in practice. Due to structural similarity and the use of shared data distributions, Q-functions often exhibit correlations and temporal coupling during training, which in turn under-

mines the effectiveness of the ensemble in uncertainty estimation and policy regularization. This limitation motivate a new thinking: Complete independence is not necessary; the key lies in how to dynamically maintain effective diversity among the Q-functions participating in updates during training through appropriate mechanisms.

In this paper, we propose *sliding window delayed gradient* (SWDG), a novel ensemble-based method that maintains diversity among Q-functions through temporal asynchrony. SWDG employs a sliding-window mechanism to construct interleaved ensembles of freshly updated and delayed critics, so that each update involves a different set of timestep-aligned Q-functions. This sliding-window schedule encourages temporal decorrelation across critics, lowers inter-critic correlation, and mitigates overfitting to recently sampled targets, thereby improving stability and generalization. Meanwhile, SWDG introduces a delay-gradient target: the minimum over delayed critics is used as the training target for the windowed ensemble, decoupling target estimation from freshly updated networks and enforcing a conservative learning signal. By separating target construction from current updates and taking a delayed minimum, SWDG suppresses optimistic bootstrapping and breaks harmful feedback loops, yielding a more reliable, lower-variance target with stronger conservative guarantees. From a computational standpoint, updating only a window of critics at each step reduces per-iteration cost and enables smaller ensembles to attain comparable performance, improving efficiency and scalability. We further provide a theoretical analysis showing how windowed asynchrony attenuates shared-update coupling and strengthens pessimistic estimation bounds. We evaluate SWDG on the standard offline RL benchmark D4RL Fu et al. (2020); the results show that SWDG significantly outperforms existing state-of-the-art methods across diverse tasks while maintaining stable, conservative value estimates.

## 2 RELATED WORK

Offline reinforcement learning has gained significant attention in recent years, as the potential of learning RL agents without interaction of the environment, which makes it relevant to real scenarios Levine et al. (2020). However, learning solely from offline data is the source of bias in the model, since the dataset may not cover the entire state–action space. Previous approaches Kumar et al. (2019); Fujimoto & Gu (2021); Ran et al. (2023) impose constraints to obtain pessimistic policy, which effectively mitigate the overestimation issue but may result in overly conservative behavior. In contrast, uncertainty estimation methods can promote efficient exploration by applying the principle of optimism with respect to the estimated uncertainties Ciosek et al. (2019).

In traditional, uncertainty estimation has been key to promote exploration policy, used to encourage agents to explore paths of high uncertainty. In contrast, offline RL requires the agent to avoid exploration and encourages it to follow paths with low uncertainty Ghasemipour et al. (2022). Both paradigms require accurate uncertainty estimates. Various methods have been developed for uncertainty estimation, including Bayesian uncertainty Yang et al. (2022a); Wu et al. (2021), bootstrapping Kostrikov & Nachum (2020), random distillation network Nikulin et al. (2023); Hu et al. (2025); Yang et al. (2024), and the ensemble-based method Agarwal et al. (2020); An et al. (2021); Chen et al. (2021); Ghasemipour et al. (2022); Bai et al. (2022); Ghosh et al. (2022). We mainly focus on the ensemble-based method. Ensemble DQN Anschel et al. (2017) reduces overestimation bias by using the average Q-value of an ensemble to compute the Bellman target. REDQ Chen et al. (2021) randomly selects a subset and takes its minimum value as the target Q-value, combining it with the update-to-data method to improve sample efficiency and control overestimation bias. EDAC An et al. (2021) uses similarity loss to promote diversity in the Q-function, thereby optimizing the variance of values in Q-ensembles on OOD data. While these methods achieve impressive results, they not only be affected by the performance of individual networks but also overlook the correlation introduced by shared, synchronized Bellman targets. We propose a sliding window mechanism to construct different temporal Q-functions, thus avoiding overfitting of the ensemble and promoting the temporal diversity.

## 3 PRELIMINARIES

### 3.1 OFFLINE REINFORCEMENT LEARNING

Reinforcement learning is defined as a Markov decision process (MDP) with five-tuple $(\mathcal{S}, \mathcal{A}, P, r, \gamma)$, where $\mathcal{S}$, $\mathcal{A}$, $P(s'|s,a)$, $r(s,a) \in \mathbb{R}$ denotes the state space, action space, transition dynamics, reward function, and discount factor, respectively. In the offline RL setting, the agent cannot directly interact with the environment. Instead, it is provided with a fixed dataset $\mathcal{D} = \{(s_i, a_i, r_i, s'_i)\}_{i=1}^{N}$, which is typically collected by an unknown behavior policy $\pi_\beta$. The goal of offline RL is to produce a new policy $\pi(a|s)$ from $\mathcal{D}$ that maximizes the expected cumulative discounted return and generalizes beyond the behavior policy and performs well when deployed.

### 3.2 ENSEMBLE Q-LEARNING

Ensemble methods construct multiple Q-functions $\{Q_k\}_{k=1}^{K}$, each trained with different random initialization, sampling, or regularization. Formally, each estimator can be written as $Q_k(s,a) = Q^*(s,a) + \epsilon_k(s,a)$, where $\epsilon_k(s,a)$ denotes the approximation error. If the errors are approximately independent across $k$, aggregating the ensemble can reduce variance. The ensemble mean $\mu(s,a) = \frac{1}{K} \sum_{k=1}^{K} Q_k(s,a)$ serves as a more stable estimate of $Q^*(s,a)$, while the variance $\sigma^2(s,a) = \frac{1}{K} \sum_{k=1}^{K} (Q_k(s,a) - \mu(s,a))^2$ reflects epistemic uncertainty.

Different ensemble aggregation strategies exploit this mean–variance trade-off: taking the mean reduces variance, while conservative estimators such as the minimum $Q_{\min}(s,a) = \min_k Q_k(s,a)$ or the lower-confidence bound $Q_{\text{LCB}}(s,a) = \mu(s,a) - \alpha\sigma(s,a)$ ($\alpha > 0$) explicitly penalize uncertainty. These ensemble techniques provide more reliable Q-value estimates, thereby improving stability and robustness in offline RL. It is noted that most of these methods share a single Bellman target across ensemble members; in contrast, the sliding window mechanism proposed in this paper decouples target dependence along the temporal and member dimensions, inherently avoiding the drawbacks of shared targets while estimating the Q-values within the window from a conservative perspective.

## 4 METHODOLOGY

In this section, we mainly introduce the motivation, overall framework, and the comprehensive algorithm. In section 4.1, we detail the motivation for SWDG. In section 4.2, we give the overall framework and calculation process of SWDG, In section 4.3, we introduce the comprehensive algorithm of SWDG.

### 4.1 MOTIVATION

Q-ensembles can reduce the overall estimation error by constructing a collection of critics. However, since extrapolation errors accumulate through the bootstrapping process, and these ensemble networks often share the same target function, these errors are often further amplified when the critics are trained synchronously. A common view is that this issue stems from the high correlation among Q-networks. Methods such as EDAC attempt to address this by introducing diversity losses to encourage differences among critics, thereby reducing correlation and improving policy performance. Nevertheless, because Q-functions are structurally homogeneous and share the same data source and training targets, their practical independence remains very limited, and the effectiveness of such methods is unsatisfactory. Moreover, these approaches fail to account for the potential overfitting risk inherent in ensembles. Given that long-term independence is almost impossible to achieve, this paper instead asks: can we ensure temporal decorrelation among Q-networks along the temporal dimension, while simultaneously addressing overfitting, so as to mitigate extrapolation error during training?

### 4.2 SLIDING WINDOW DELAYED GRADIENT

To reduce the correlation between Q-networks and avoid overfitting of ensembles, we propose sliding window delayed gradient (SWDG). Instead of synchronously training all $N$ critics $\{Q_{\phi_i}\}_{i=1}^{N}$,

our method activates a contiguous subset of window size $m < N$ at each iteration and advances the window with a step size $k$. Formally, the active set is defined as Eq. 1.

$$\mathcal{W}_t = \{(j_t + p) \bmod N \mid p = 0, 1, \ldots, m - 1\}, \tag{1}$$

where $\mathcal{W}_t$ denotes the active window of critics at iteration $t$, with window size $m$ and starting index $j_t \in \{0, k, 2k, \ldots\} \bmod N$ that shifts forward by a fixed step size $k$.

Compared with conventional ensemble-based methods, this design provides two direct benefits. By updating only a subset of critics at each iteration, the gradients received by different networks are naturally de-synchronized, which lowers the instantaneous correlation among their parameters and mitigates the tendency of the ensemble to collapse into highly similar solutions. At the same time, restricting updates to a window of size $m$ instead of the full ensemble size $N$ significantly reduces the per-iteration computational burden, making the method more efficient and scalable without sacrificing the representational richness of the ensemble.

The number of newly entered critics when moving from $\mathcal{W}_{t-1}$ to $\mathcal{W}_t$ is

$$U_t := \mathcal{W}_t \setminus \mathcal{W}_{t-1}, \qquad |U_t| = u = \min\{\delta, N - \delta, m\}, \tag{2}$$

which exactly characterizes the overlap between successive windows. This temporal overlap induces a natural *age structure* within the active window. Let $\mathrm{age}_i(t)$ denote the number of iterations since critic $i$ was last updated. The empirical age frequencies inside the window are

$$w_{\ell,t} := \frac{1}{m} \left| \{ j \in \mathcal{W}_t : \mathrm{age}_j(t) = \ell \} \right|, \qquad \sum_{\ell \geq 1} w_{\ell,t} = 1. \tag{3}$$

This age structure provides a probabilistic view of critic freshness: $w_{1,t}$ indicates the fraction of just-updated critics, while higher ages correspond to critics that retain older parameter states. In practice, this creates a stochastic smoothing effect across updates, because the target computation at iteration $t$ depends on a mixture of fresh and stale critics. Such smoothing prevents abrupt target shifts and alleviates variance accumulation.

When $0 < k < m$, only ages 1 and 2 appear and Eq. 3 simplifies to

$$w_{1,t} = \frac{m - u}{m}, \qquad w_{2,t} = \frac{u}{m}. \tag{4}$$

The windowed aggregation is then used to construct both the policy improvement objective and the critic target. For a replay transition $(s, a, r, s')$, we define

$$\mathcal{L}_{\mathrm{actor}}(\theta) = \frac{1}{|B_t|} \sum_{s \in B_t} \left( \min_{j \in \mathcal{W}_t} Q_{\phi_j}(s, \tilde{a}_\theta(s)) - \beta \log \pi_\theta(\tilde{a}_\theta(s) \mid s) \right). \tag{5}$$

We then form a delayed-gradient target by taking the minimum *only over* $U_t$:

$$y_t(r, s') = r + \gamma \left( \min_{j \in U_t} Q_{\phi'_j}(s', a') - \beta \log \pi_\theta(a' \mid s') \right), \qquad a' \sim \pi_\theta(\cdot \mid s'). \tag{6}$$

Note that restricting the target computation to $U_t$ further emphasizes the role of the most recently updated critics. This selective pessimism avoids "double counting" the stale critics when forming the target, but still leverages them in the actor loss. Consequently, the actor is discouraged by the full window, while the critic update is guided only by the fresh subset. This asymmetry strengthens regularization: the policy is optimized conservatively, while critics evolve in a gradually decorrelated manner.

This construction is not merely "using fewer critics." The temporal movement creates a mixture of *new* critics (just updated) and *old* critics (not updated in the previous step) inside $\mathcal{W}_t$. The old critics are less exposed to the latest update noise and, distributionally, remain closer to the behavior policy; combining them with the new critics in the windowed minimum preserves OOD pessimism while preventing a single freshly-updated low estimate from dominating the target. From an optimization viewpoint, the age mixture implies that not all active critics are driven by the same instantaneous

gradient, which weakens the shared Bellman drive and reduces temporal correlation among critics. Meanwhile, the minimum is restricted to $m$ critics, so the extremal tail amplification depends on $m$ rather than $N$, mitigating systematic over-conservatism. As the window slides, every critic is periodically included, which prevents bias from a fixed subset and acts as a temporal regularizer: the reduced co-activation and age-induced decorrelation mitigate overfitting. Computation is also reduced, since each iteration calculates only $m$ critics.

Figure 1: Illustration example of SWDG. The left part illustrates the overall process from input to the output Q-values. The right part provides a detailed view of the sliding window computation.

The framework of SWDG is shown in Fig. 1. On the left, an input batch is encoded by a shared backbone and fed to a Q-ensemble; a sliding window (orange) selects the active critics for this step and then shifts along the ensemble over time so different critics participate in successive updates. On the right, within the window we impose the EDAC cosine diversity loss to decorrelate critics. Targets are computed from a delayed subset of critics (delayed-min), while the active critics use these targets for regression—separating target construction (delayed) from parameter updates (current). The overall training procedure is presented in Algorithm 1.

### 4.3 THEORETICAL ANALYSIS.

To analyze the effect of contiguous sliding window updates on both pessimistic estimation and temporal decorrelation in critic ensembles, we will begin by establishing some additional minimum sufficient assumptions, then we derived a pessimistic lower bound for the sliding window mechanism and an upper bound for temporal decorrelation. Finally, we demonstrate that the delayed-gradient target can strengthen this upper bound.

**Assumption 4.1** (Bounded values & discount). *The reward is bounded and the MDP is discounted:* $|r| \leq R_{\max}$ *and* $\gamma \in (0, 1)$. *Consequently, there exists* $Q_{\max} < \infty$ *such that for any policy* $\pi$, $|Q^\pi(s, a)| \leq Q_{\max}$; *we assume the learned critics satisfy* $|Q_{\phi_i}(s, a)| \leq Q_{\max}$ *on the support of the evaluation distribution.*

**Lemma 4.2** (Quantile bound under boundedness). *Under Assumption 4.1, for any critic $i$, any $(s, a)$ in the evaluation support, and any $\delta \in (0, 1/e)$,*

$$\Pr\big(Q_{\phi_i}(s, a) - Q^\pi(s, a) \ \leq \ -K \ln(1/\delta)\big) \ \leq \ \delta \qquad \text{for any } K > 2Q_{\max}.$$

**Assumption 4.3** (Contiguous sliding window and coverage). *The window $\mathcal{W}_t$ is contiguous as above. Either the start index $j_t$ is uniform on $\{1, \ldots, N\}$, or statistics are averaged over a full cycle when $\gcd(s, N) = 1$.*

---

**Algorithm 1** Sliding Window Delayed Gradient (SWDG)

---

1: Initialize policy parameters $\theta$, Q-function parameters $\{\phi_j\}_{j=1}^N$, target Q-function parameters $\{\phi'_j\}_{j=1}^N$, window size $w$, slide step $k$, and offline replay buffer $\mathcal{D}$
2: Initialize window start index $j \leftarrow 0$
3: **while** not converged **do**
4:     Sample a mini-batch $B = \{(s, a, r, s')\}$ from $\mathcal{D}$
5:     Assign active window $W_t \leftarrow \{j_t, j_t+1, \ldots, j_t+w-1\} \pmod{N}$, and delayed window $U_t \leftarrow W_{t-k}$
6:     Sample a mini-batch $B_t = \{(s, a, r, s')\}$ from $\mathcal{D}$
7:     Compute target shared by critics in $\mathcal{W}$:

$$y_t(r, s') = r + \gamma \Big( \min_{j \in U_t} Q_{\phi'_j}(s', a') - \beta \log \pi_\theta(a' \mid s') \Big), \quad a' \sim \pi_\theta(\cdot \mid s')$$

8:     Update Q-function $Q_{\phi_i}$ (only $i \in W_t$) with gradient descent:

$$\nabla_{\phi_i} \frac{1}{|B|} \sum_{(s,a,r,s') \in B} \Big( \big(Q_{\phi_i}(s,a) - y(r,s')\big)^2 + \frac{\eta}{m-1} \sum_{1 \le i \ne j \le m} \mathrm{ES}_{\phi_i, \phi_j}(s,a) \Big)$$

9:     Update policy with gradient ascent:

$$\nabla_\theta \frac{1}{|B_t|} \sum_{s \in B_t} \Big( \min_{j \in \mathcal{W}_t} Q_{\phi_j}\big(s, \tilde{a}_\theta(s)\big) - \beta \log \pi_\theta\big(\tilde{a}_\theta(s) \mid s\big) \Big)$$

where $\tilde{a}_\theta(s)$ is sampled via $\pi(\cdot|s)$.
10:     Update target networks: $\phi'_i \leftarrow \rho \phi'_i + (1-\rho)\phi_i$
11:     **(Slide the window)** $j \leftarrow (j+k) \bmod N + 1$
12: **end while**

---

**Assumption 4.4** (Temporal attenuation of shared influence). *There exists $\theta \in (0, 1]$ such that the influence (in sup-norm) of a change made to the common target $\ell$ steps ago is attenuated by at most $\theta^\ell$. (In discounted RL, one may take $\theta = \gamma$.)*

**Lemma 4.5** (Co-activation under contiguous window). *Under Assumption 4.3, for a fixed pair $(i, k)$ with circular distance $d(i, k) \in \{1, \ldots, N-1\}$,*

$$p_{\mathrm{co}}(i, k) = \Pr[i, k \in \mathcal{W}_t] = \max\Big\{ \frac{m - d(i,k)}{N}, 0 \Big\}.$$

*Averaging over an unordered pair drawn uniformly from $\{1, \ldots, N\}$ yields the mean co-activation rate*

$$\bar{p}_{\mathrm{co}}(m, N) = \frac{\binom{m}{2}}{\binom{N}{2}}.$$

**Lemma 4.6** (Age-mixed attenuation). *Under Assumption 4.4, letting $w_{\ell,t}$ be the fraction of age-$\ell$ critics in $\mathcal{W}_t$,*

$$c_t = \sum_{\ell \ge 1} w_{\ell,t}\, \theta^\ell \le \theta, \qquad \text{and } c_t < \theta \text{ whenever } \sum_{\ell \ge 2} w_{\ell,t} > 0.$$

**Proposition 4.7** (Windowed pessimistic bound). *Under Assumption 4.1, the quantile bound in Lemma 4.2 holds uniformly on the evaluation support. Then for any $\delta \in (0, 1/e)$ and fixed $(s, a)$,*

$$\Pr\Big[ \min_{j \in \mathcal{W}_t} Q_{\phi_j}(s, a) \ge Q^\pi(s, a) - K \ln \tfrac{m}{\delta} \Big] \ge 1 - \delta,$$

*and analogously with $m$ replaced by $N$ for the full ensemble.*

**Corollary 4.8** (One-step improvement guarantee). *Define $F(\pi) = \mathbb{E}_{s \sim \mu,\, a \sim \pi}[Q^\pi(s, a) - \beta \log \pi(a|s)]$ and $L_m(\pi) = \mathbb{E}_{s \sim \mu,\, a \sim \pi}[\min_{j \in \mathcal{W}_t} Q_{\phi_j}(s, a) - \beta \log \pi(a|s)]$. Let $\varepsilon_m := K \ln \frac{m}{\delta}$ with $K > 2Q_{\max}$. Then, with probability at least $1 - \delta$ Bartlett & Mendelson (2002), simultaneously for all $\pi$,*

$$F(\pi) \le L_m(\pi) + \varepsilon_m.$$

Table 1: Performance comparison on D4RL Gym tasks, averaged over 5 random seeds. The results of CQL and IQL are taken from the EDAC An et al. (2021) paper.The EQL, ACTIVE, EDAC, RORL scores are token from the original paper. Best results per row are highlighted in bold.

| Task Name | IQL | CQL | EQL | ACTIVE | EDAC | RORL | SWDG (Ours) |
|---|---|---|---|---|---|---|---|
| halfcheetah-random | 13.1 | 35.4 | – | – | 28.4 | 28.5 | **35.8** ±**1.2** |
| halfcheetah-medium | 47.4 | 44.4 | 54.9 | 52.3 | 65.9 | 66.8 | **74.5** ± **0.5** |
| halfcheetah-expert | 95.0 | 104.8 | – | – | 106.8 | 105.2 | **107.6** ± **2.6** |
| halfcheetah-medium-expert | 86.7 | 62.4 | 90.3 | 92.9 | 106.3 | 107.8 | **109.4**±**2.2** |
| halfcheetah-medium-replay | 44.2 | 46.2 | 57.0 | 51.7 | 61.3 | 61.9 | **69.7**±**1.4** |
| halfcheetah-full-replay | – | – | – | – | 84.6 | – | **86.3**±**0.6** |
| hopper-random | 7.9 | 10.8 | – | – | 25.3 | 31.4 | **33.6**±**6.5** |
| hopper-medium | 66.2 | 86.6 | 94.2 | 86.1 | 101.6 | **104.8** | 103.8±0.4 |
| hopper-expert | 109.4 | 109.9 | – | – | 110.1 | **112.8** | 110.6±0.6 |
| hopper-medium-expert | 91.5 | **111.0** | 111.9 | 109.9 | 110.7 | **112.7** | 112.2±0.2 |
| hopper-medium-replay | 94.7 | 48.6 | 102.7 | 102.8 | 101.0 | 102.8 | **104.8**±**1.2** |
| hopper-full-replay | – | – | – | – | 105.4 | – | **109.5**±**0.7** |
| walker2d-random | 5.4 | 7.0 | – | – | 16.6 | 21.4 | **24.8**±**5.5** |
| walker2d-medium | 78.3 | 74.5 | 92.5 | 87.2 | 92.5 | **102.4** | 95.4±0.6 |
| walker2d-expert | 109.9 | **121.6** | – | – | 115.1 | 115.4 | **117.4**±**0.9** |
| walker2d-medium-expert | 109.6 | 98.7 | 111.2 | 111.7 | 114.7 | **121.2** | 115.3±0.6 |
| walker2d-medium-replay | 73.8 | 32.6 | 94.2 | 79.3 | 87.1 | 90.4 | **95.8**±**1.6** |
| walker2d-full-replay | – | – | – | – | 99.8 | – | **108.9**±**1.7** |

*In particular, for any $\pi_t^{(m)} \in \arg\max_\pi L_m(\pi)$,*

$$F(\pi_t^{(m)}) \geq \sup_\pi F(\pi) - 2\varepsilon_m.$$

**Proposition 4.9** (temporal decorrelation under contiguous window). *Let $\mathcal{C}_t$ be a normalized mean pairwise correlation among updates of active critics, and $\mathcal{C}_t^{(\text{base})}$ its counterpart when all $N$ critics are updated synchronously. Under Assumptions 4.3–4.4. Assume further a shared-influence decomposition of per-critic updates at step $t$: $\Delta_{i,t} = a_{i,t} S_t + \eta_{i,t}$, where $|a_{i,t}| \leq c_t$, $S_t$ is a common term (driven by the target), and $\eta_{i,t}$ are idiosyncratic terms whose cross-covariances vanish across different critics at the same step. Then, using the averaged co-activation rate $\bar{p}_{\text{co}}(m, N) = \binom{m}{2}/\binom{N}{2}$,*

$$\mathcal{C}_t \leq \bar{p}_{\text{co}}(m, N) c_t^2 \mathcal{C}_t^{(\text{base})} \leq \bar{p}_{\text{co}}(m, N) \theta^2 \mathcal{C}_t^{(\text{base})}.$$

*In particular, if $m < N$ and $\sum_{\ell \geq 2} w_{\ell,t} > 0$, then $\mathcal{C}_t < \mathcal{C}_t^{(\text{base})}$ Polyak & Juditsky (1992).*

*Remark* 4.10 (Delayed-gradient strengthens decorrelation). If the target is evaluated on $U_t = \mathcal{W}_t \setminus \mathcal{W}_{t-1}$ while gradients update all members of $\mathcal{W}_t$, the target's age histogram concentrates on $\ell \geq 2$, which further reduces $c_t$ and tightens Proposition 4.9.

## 5 EXPERIMENTS

In this section, we present empirical evaluations of SWDG against baseline algorithms. We first provide a comparison results with baseline model-free offline RL methods on the D4RL benchmark Fu et al. (2020). We then analyze the effect of diversity loss and sliding window parameters in ablation study.

### 5.1 COMPARISONS AND BASELINE

We first evaluate our method on the D4RL Fu et al. (2020) MuJoCo Gym domain, where we choose Halfcheetah, Hopper, and Walker2d as tasks. In these tasks, we use six different offline datasets including "medium", "medium-replay", "medium-expert", "expert", "full-replay", and "random", each collected by different behavior policies. Additionally, We conduct our method on Adroit domains which consist of 4 different tasks with "human" and "cloned" datasets.

Table 2: Normalized average returns on D4RL Adroit tasks, averaged over 5 random seeds.

| Task Name | BC | SAC | REM | CQL | EDAC | SWDG (ours) |
|---|---|---|---|---|---|---|
| pen-human | $25.8 \pm 8.8$ | $4.3 \pm 3.8$ | $5.4 \pm 4.3$ | 55.8 | $52.1 \pm 8.6$ | **65.4±7.2** |
| hammer-human | $3.1 \pm 3.2$ | $0.2 \pm 0.0$ | $0.3 \pm 0.0$ | 2.1 | $0.8 \pm 0.4$ | 1.0±0.2 |
| door-human | $2.8 \pm 0.7$ | $-0.3 \pm 0.0$ | $-0.3 \pm 0.0$ | 9.1 | $10.7 \pm 6.8$ | 7.6±2.5 |
| relocate-human | $0.0 \pm 0.0$ | $-0.3 \pm 0.0$ | $-0.3 \pm 0.0$ | 0.35 | $0.1 \pm 0.1$ | 0.1 ± 0.1 |
| pen-cloned | $38.3 \pm 11.9$ | $-0.8 \pm 3.2$ | $-1.0 \pm 0.1$ | 40.3 | $68.2 \pm 7.3$ | 76.2 ± 7.3 |
| hammer-cloned | $0.7 \pm 0.3$ | $0.1 \pm 0.1$ | $-0.3 \pm 0.0$ | 5.7 | $0.3 \pm 0.0$ | 0.3 ± 0.0 |
| door-cloned | $0.0 \pm 0.0$ | $-0.3 \pm 0.1$ | $-0.3 \pm 0.0$ | 3.5 | $9.6 \pm 8.3$ | 8.2±5.7 |
| relocate-cloned | $0.1 \pm 0.0$ | $-0.1 \pm 0.1$ | $-0.2 \pm 0.2$ | -0.1 | $0.0 \pm 0.0$ | 0.0 ± 0.0 |

We compare our proposed SWDG with remarkable ensemble-based algorithm, including EDAC An et al. (2021), RORLYang et al. (2022b), ACTIVE Chen et al. (2025), and Zhang et al. (2024), and 2 classical model-free offline RL method IQL Kostrikov et al. (2021) and CQL Kumar et al. (2020). For the baseline, we directly report the normalized score from papers of prior methods, and we choose EDAC An et al. (2021) as the basic methods for SWDG. The results are shown in Table 1. In the MuJoCo domain, SWDG surpasses existing baselines on the majority of datasets, demonstrating strong stability and cross-environment consistency. The advantage is especially evident on the medium and medium-replay datasets, where SWDG outperforms compared approaches by a wide margin, indicating stronger robustness to data quality.

The evaluation results are reported in Table 2. On the pen tasks, SWDG demonstrates performance that is comparable to or better than the previous state-of-the-art methods. In particular, for the pen-human setting, SWDG attains approximately 25% higher performance compared to EDAC, highlighting its clear advantage. These findings suggest that the proposed approach is capable of effectively addressing complex control problems that involve high-dimensional state and action spaces.

## 5.2 ABLATION STUDY

The setting of the window size is crucial to the SWDG. It not only affects the final outcome of the algorithm but also impacts the computational time of the entire method. Therefore, we conducted an ablation study on the window size. We conduct the study on the hopper task with 3 datasets, and the percentage refers to the proportion of the window size relative to the total number of Q-ensembles units. As shown in Fig. 2. The results indicate that introducing a sliding window consistently improves performance without degrading stability. Moreover, smaller window sizes tend to yield better returns, indicating that stronger temporal asynchrony helps maintain critic diversity while reducing computational overhead per step.

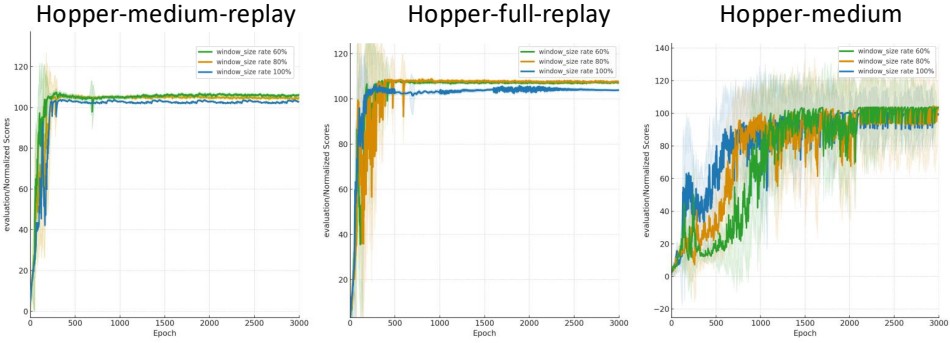

Figure 2: Normalized score of SWDG with different window size.

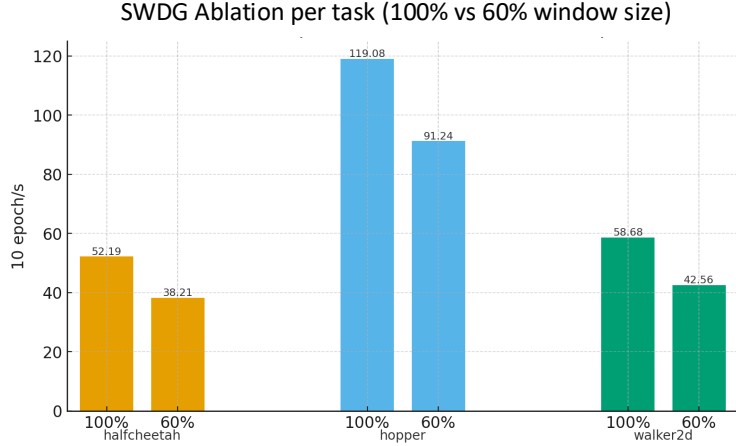

Figure 3: Runtime of SWDG for different window sizes over 10 epochs

We further compare runtime across different window sizes. As shown in Fig. 3 The results show that replacing the full Q-ensembles $N$ with a sliding update of window size $m$ reduces the per-step computation from $O(N)$ to $O(m)$, thereby markedly lowering running time.

The delayed gradient objective introduces additional robustness during optimization, preventing target value shifts caused by large errors in individual Q-networks. We conducted ablation experiments to evaluate the effectiveness of this mechanism. As shown in the table 3, the delayed gradient objective significantly improves policy performance in datasets like medium and medium-replay. In these datasets, the delayed-gradient target can provide around a $5\%$ performance improvement. This is because the mechanism provides a dynamic objective domain, enhancing the robust lower bound of the value function in these non-expert environments and preventing overly pessimistic value estimates.

Table 3: Comparative results of the with/without delay gradient target, obtained from 5 random seeds.

| Tasks | Without | **With** |
|---|---|---|
| halfcheetah-medium | $68.7 \pm 0.3$ | $\mathbf{74.5 \pm 0.5}$ |
| halfcheetah-medium-replay | $64.3 \pm 1.2$ | $\mathbf{69.7 \pm 1.4}$ |
| halfcheetah-medium-expert | $108.2 \pm 1.8$ | $\mathbf{109.4 \pm 2.2}$ |
| hopper-medium | $103.4 \pm 0.4$ | $103.8 \pm 0.4$ |
| hopper-medium-replay | $102.7 \pm 1.4$ | $\mathbf{104.8 \pm 1.2}$ |
| hopper-medium-expert | $112.0 \pm 0,2$ | $112.2 \pm 0.2$ |
| walker2d-medium | $94.7 \pm 0.5$ | $95.4 \pm 0.6$ |
| walker2d-medium-replay | $87.1 \pm 1.8$ | $\mathbf{95.8 \pm 1.6}$ |
| walker2d-medium-expert | $114.7 \pm 0.4$ | $115.3 \pm 0.6$ |
| average score | $95.08$ | $97.87$ |

## 6 CONCLUSION

In this work, we propose Sliding Window Delayed Gradient (SWDG), which updates only a moving window of critics and forms targets from the freshly updated subset. This decorrelates the ensemble, curbs overfitting while preserving pessimism, and reduces runtime. We theoretically demonstrated the effectiveness of SWDG and experimentally validated this theory. The results show that SWDG exhibits state-of-the-art performance on various datasets. We will also focus on researching the extension of sliding window mechanisms to conventional non-ensemble methods in the future.

## ETHICS STATEMENT

This work adheres to the ICLR Code of Ethics. In this study, no human subjects or animal experimentation was involved. All datasets used, including D4RL Fu et al. (2020), were sourced in compliance with relevant usage guidelines, ensuring no violation of privacy. We have taken care to avoid any biases or discriminatory outcomes in our research process. No personally identifiable information was used, and no experiments were conducted that could raise privacy or security concerns. We are committed to maintaining transparency and integrity throughout the research process.

## REPRODUCIBILITY STATEMENT

We have made every effort to ensure that the results presented in this paper are reproducible. All code and datasets have been made publicly available in an anonymous repository to facilitate replication and verification. The experimental setup, including training steps, model configurations, and hardware details, is described in detail in the paper.

Additionally, the D4RL datasets are publicly available, ensuring consistent and reproducible evaluation results.

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

## A  APPENDIX

### A. PROOFS

**Lemma 4.2:** By Assumption 4.1, for any $(s, a)$ on the evaluation support, $|Q^\pi(s, a)| \leq Q_{\max}$ and $|Q_{\phi_i}(s, a)| \leq Q_{\max}$, hence

$$-2Q_{\max} \leq Q_{\phi_i}(s, a) - Q^\pi(s, a) \leq 2Q_{\max}.$$

Fix $\delta \in (0, 1/e)$ so that $\ln(1/\delta) \geq 1$. For any $K > 2Q_{\max}$ we have $K \ln(1/\delta) > 2Q_{\max}$, thus the event $\{ Q_{\phi_i}(s, a) - Q^\pi(s, a) \leq -K \ln(1/\delta) \}$ is empty, and its probability is $\leq \delta$.  □

**Lemma 4.5:** View $\{1, \ldots, N\}$ as a ring. Fix $(i, k)$ with circular distance $d = d(i, k) \in \{1, \ldots, N-1\}$. For start index $j$, the active window is $\mathcal{W}_t(j) = \{j, \ldots, j + m - 1\} \pmod{N}$. The pair $(i, k)$ is co-activated iff the length-$m$ arc covers their shortest arc, which is possible iff $m > d$. When $m > d$, exactly $m - d$ start indices $j$ yield co-activation; under uniform $j$ (or cycle averaging),

$$p_{\text{co}}(i, k) = \frac{\max\{m - d, \, 0\}}{N}.$$

Averaging over a uniformly drawn unordered pair gives $\bar{p}_{\text{co}}(m, N) = \binom{m}{2}/\binom{N}{2}$, since a window contains $\binom{m}{2}$ in-window pairs among $\binom{N}{2}$ total.  □

**Lemma 4.6:** Let $w_{\ell,t}$ be the fraction of age-$\ell$ critics in $\mathcal{W}_t$. With $\theta \in (0, 1]$ and $\sum_{\ell \geq 1} w_{\ell,t} = 1$,

$$c_t = \sum_{\ell \geq 1} w_{\ell,t} \theta^\ell \leq \sum_{\ell \geq 1} w_{\ell,t} \theta = \theta.$$

If $\sum_{\ell \geq 2} w_{\ell,t} > 0$ and $\theta < 1$, then some terms use $\theta^\ell \leq \theta^2 < \theta$, hence $c_t < \theta$.  □

**Proof of Proposition 4.7:** By Lemma 4.2 (uniform on the evaluation support), for any $j \in \mathcal{W}_t$ and $\delta' \in (0, 1/e)$,

$$\Pr\Big(Q_{\phi_j}(s, a) - Q^\pi(s, a) \leq -K \ln(1/\delta')\Big) \leq \delta', \qquad K > 2Q_{\max}.$$

Set $\delta' = \delta/m$. Union bound over $j \in \mathcal{W}_t$ yields

$$\Pr\Big(\exists j \in \mathcal{W}_t : \, Q_{\phi_j}(s, a) - Q^\pi(s, a) \leq -K \ln \frac{m}{\delta}\Big) \leq \sum_{j \in \mathcal{W}_t} \frac{\delta}{m} = \delta.$$

Equivalently, with probability at least $1 - \delta$,

$$\min_{j \in \mathcal{W}_t} Q_{\phi_j}(s, a) \geq Q^\pi(s, a) - K \ln \frac{m}{\delta}.$$

Replacing $m$ by $N$ gives the full-ensemble variant.  □

**Corollary 4.8:** By Proposition 4.7 (uniform version on the evaluation support) and for any $\pi$,

$$\min_{j \in \mathcal{W}_t} Q_{\phi_j}(s, a) \geq Q^\pi(s, a) - \varepsilon_m, \qquad \varepsilon_m := K \ln \frac{m}{\delta}.$$

Taking expectations over $s \sim \mu$ and $a \sim \pi$, and subtracting the same entropy term,

$$L_m(\pi) \geq F(\pi) - \varepsilon_m \implies F(\pi) \leq L_m(\pi) + \varepsilon_m.$$

Let $\pi_t^{(m)} \in \arg\max_\pi L_m(\pi)$. Then

$$\sup_\pi F(\pi) \leq \sup_\pi \big(L_m(\pi) + \varepsilon_m\big) = L_m(\pi_t^{(m)}) + \varepsilon_m \leq F(\pi_t^{(m)}) + 2\varepsilon_m,$$

which rearranges to the stated bound. *Uniformity note:* a standard $\epsilon$-net/Rademacher-complexity argument (e.g., Bartlett & Mendelson (2002)) promotes pointwise to uniform control.  □

**Proof of Proposition 4.9:** Assume the shared-influence model $\Delta_{i,t} = a_{i,t} S_t + \eta_{i,t}$ with $|a_{i,t}| \leq c_t$ and $\text{Cov}(\eta_{i,t}, \eta_{k,t}) = 0$ for $i \neq k$. For a co-activated pair $(i, k)$,

$$\text{Cov}(\Delta_{i,t}, \Delta_{k,t}) = \text{Cov}(a_{i,t} S_t + \eta_{i,t},\ a_{k,t} S_t + \eta_{k,t}) = a_{i,t} a_{k,t} \,\text{Var}(S_t).$$

Normalizing to correlation coefficients gives $|\rho_{i,k}^{(\text{act})}| \leq c_t^2 |\rho_{i,k}^{(\text{base})}|$. Only co-activated pairs contribute under the windowed update; by Lemma 4.5, the mean contributing fraction is $\bar{p}_{\text{co}}(m, N) = \binom{m}{2}/\binom{N}{2}$. Therefore

$$\mathcal{C}_t \ \leq \ \bar{p}_{\text{co}}(m, N)\, c_t^2\, \mathcal{C}_t^{(\text{base})} \ \leq \ \bar{p}_{\text{co}}(m, N)\, \theta^2\, \mathcal{C}_t^{(\text{base})},$$

where the last inequality uses Lemma 4.6. If $m < N$ then $\bar{p}_{\text{co}}(m, N) < 1$; if also $\sum_{\ell \geq 2} w_{\ell,t} > 0$ and $\theta < 1$, then $c_t < \theta \leq 1$, hence $\mathcal{C}_t < \mathcal{C}_t^{(\text{base})}$. $\qquad\square$

**Remark 4.10:** When the target is evaluated on $U_t = \mathcal{W}_t \backslash \mathcal{W}_{t-1}$ while gradients update all members of $\mathcal{W}_t$, the critics contributing to the target are newly entered and were typically not updated at $t-1$. Thus their ages satisfy $\ell \geq 2$ at evaluation time, shifting the age histogram towards larger $\ell$. By Lemma 4.6, this reduces $c_t = \sum_{\ell \geq 1} w_{\ell,t} \theta^\ell$, which tightens the bound in Proposition 4.9. (Alternatively, a target network with update lag achieves the same effect.) $\qquad\square$

B. IMPLEMENTATION DETAILS

Our approach uses EDAC as the baseline. To isolate the contribution of our method to ensemble performance, we keep all standard hyperparameters aligned with EDAC An et al. (2021), except for two newly introduced sliding window coverage and the window step. The specific configurations are provided in Tab. 4.

Table 4: Hyperparameters used in the D4RL MuJoCo Gym experiments.

| Task Name | Q ensemlbes | Widow(rate, step) |
|---|---|---|
| halfcheetah-random | 6 | 0.6, 2 |
| halfcheetah-medium | 6 | 0.6, 2 |
| halfcheetah-expert | 10 | 0.6, 4 |
| halfcheetah-medium-expert | 10 | 0.6, 4 |
| halfcheetah-medium-replay | 6 | 0.6, 2 |
| halfcheetah-full-replay | 10 | 0.6, 4 |
| hopper-random | 50 | 0.6, 10 |
| hopper-medium | 50 | 0.6, 10 |
| hopper-expert | 50 | 0.8, 20 |
| hopper-medium-expert | 50 | 0.8, 20 |
| hopper-medium-replay | 50 | 0.6, 10 |
| hopper-full-replay | 50 | 0.6, 10 |
| walker2d-random | 10 | 0.6, 4 |
| walker2d-medium | 10 | 0.6, 4 |
| walker2d-expert | 10 | 0.8, 4 |
| walker2d-medium-expert | 10 | 0.6, 4 |
| walker2d-medium-replay | 10 | 0.6, 4 |
| walker2d-full-replay | 10 | 0.6, 4 |

**Hardware** Since the hardware is relevant to the runtime in our experiments, we specify the hardware we used as follows:

- NVIDIA RTX 4090
- Intel(R) Xeon(R) Platinum 8352V CPU

## THE USE OF LARGE LANGUAGE MODELS (LLMS)

Large Language Models (LLMs) were used to aid in the writing and polishing of the manuscript. Specifically, we used an LLM to assist in refining the language, improving readability, and ensuring clarity in various sections of the paper. The model helped with tasks such as sentence rephrasing, grammar checking, and enhancing the overall flow of the text.

It is important to note that the LLM was not involved in the ideation, research methodology, or experimental design. All research concepts, ideas, and analyses were developed and conducted by the authors. The contributions of the LLM were solely focused on improving the linguistic quality of the paper, with no involvement in the scientific content or data analysis.

The authors take full responsibility for the content of the manuscript, including any text generated or polished by the LLM. We have ensured that the LLM-generated text adheres to ethical guidelines and does not contribute to plagiarism or scientific misconduct.

