# OpenReview forum: "Sliding Window–Based Q-Ensemble for Offline Reinforcement Learning"
_ICLR.cc/2026/Conference — ICLR 2026 Conference Withdrawn Submission_

### Official Review · Reviewer_LF2L · 2025-10-29

**Soundness:** 1
**Presentation:** 1
**Contribution:** 1
**Rating:** 2
**Confidence:** 4

**Summary:**

The paper considers offline reinforcement learning and proposes to decorrelate Q-ensembles through a sliding window approach. The claimed contributions include state-of-the-art results on D4RL, backed up with theoretical results.

**Strengths:**

- The observation that the functions in the Q-ensemble are correlated is reasonably well-motivated and makes intuitive sense,
- the proposed fix is very simple to implement - which I consider a merit,
- the result on D4RL appear to be clear improvements over the baselines.

**Weaknesses:**

**W1. Disconnect between claims and evidence**

There is a fundamental discrepancy between what is claimed and the (experimental or theoretical) evidence. To exemplify:
- The motivation for the proposed method is that Q-ensembles "often exhibit correlations and temporal coupling during training" - but this claim is not backed up by any evidence.
- "SWDG suppresses optimistic bootstrapping and break harmful feedback loops, yielding a more reliable, lower-variance target with stronger conservative guarantees". There are indeed theoretical results on the conservative guarantees (although I have my qualms about those, see W2) but no other parts of this claim are experimentally verified even though I think it would be feasible to do so.
- "SWDG significantly outperforms existing state-of-the-art methods across diverse tasks while maintaining stable, conservative value estimates". Nowhere is it shown that SWDG maintains stable, conservative value estimates.

**W2. Methodology hard to follow**

The sections on Related work and Preliminaries give a relatively high-level overview of ensemble Q-learning. Then, the Methodology section jumps into what is *new*, but there is a clear gap in the presentation where you would normally expect the standard method to be presented in enough detail to clearly contrast it against the proposed method. I strongly suggest to revise the Preliminaries to include such a presentation. Some examples of things warranting explanation:
- "the variance reflects epistemic uncertainty" (isn't it rather aleatoric?)
- "most of these methods share a single Bellman target",
- "extrapolation errors accumulate through the bootstrapping process" (what bootstrapping process?),
- "critics are trained synchronously",
- "Q-functions are structurally homogenous and share the same data source and training targets",
- "long-term independence is almost impossible to achieve"

**W3. Theoretical results incomplete**

The theoretical analysis in Section 4.3 is poorly structured: it is just a sequence of assumptions and lemmas/proposition without motivation, proof, or interpretation. Moreover, they seem incomplete. Consider, e.g., Lemma 4.2: since the left-hand side is independent of $\delta$ you could take $\delta\to 0^+$, which would, I think, imply that $Q_\phi\to Q$. But that doesn't seem correct. However, since there is no proof, I can't verify the reasoning.

Minor things:
- l.92 "In traditional", missing "RL".
- l.113 missing $\gamma$.
- l.123  "If the errors are approximately independent across k, aggregating the ensemble can reduce variance". This is true even if they're correlated.
- l.130 I suppose $Q_{LCB}$ assumes Gaussianity. Why does that hold?
- In algorithm 1, you say that you do gradient descent, but state just the gradient.
- Table 2: why different baselines than in Table 1?

**Questions:**

To change my opinion, the required changes would be too extensive to be accomplished during a rebuttal period.

---

### Official Review · Reviewer_qddo · 2025-10-30

**Soundness:** 3
**Presentation:** 2
**Contribution:** 1
**Rating:** 2
**Confidence:** 4

**Summary:**

The paper introduces the Sliding Window Delayed Gradient (SWDG) method, an ensemble approach designed to enhance the independence and diversity of Q-functions in Offline Reinforcement Learning. SWDG addresses a key limitation of existing ensemble methods—the loss of functional independence—by utilizing a sliding window mechanism for delayed gradient updates. The paper provides a mathematical analysis intended to support the training methodology and evaluate the approach empirically on the D4RL benchmark.

**Strengths:**

1. The core idea of decoupling the gradient updates across an ensemble using a sliding window is novel and a promising direction for resolving the functional dependence issue in Q-ensemble methods.
2. The paper includes a mathematical analysis intended to support the mechanism of the proposed SWDG method.

**Weaknesses:**

1. The results of experiments are not convincing. Although SWDG achieves a better average score compared to baselines, the improvements lack statistical significance across the majority of the D4RL tasks reported.
2. The paper contains several ambiguities and unexplained notation. For instance, the variable $\delta$ in Equation (2) is undefined. The definitions and roles of the sliding window variables should be made explicit in the main text.

**Questions:**

Questions
1. In Table 2,What specific characteristics of the 'pen-human' dataset or task dynamic might explain this significant performance gain, and why is this single result considered sufficient evidence that SWDG is superior to other methods, given its comparable or worse performance on the other seven tasks?

Suggestions
1. It is hard to see the legend and axis labels in Figure 2. It would be better to increase the font size.
2. In Line 168, $j_t = kt \mod N$ would be better to understand.
3. In Algorithm 1, Line 4 and Line 6 are duplicated. One of the lines should be removed.
4. In Algorithm 1, Line 11 will move $j$ by $k+1$, +1 should be removed here.

---

### Official Review · Reviewer_GARm · 2025-11-01

**Soundness:** 3
**Presentation:** 2
**Contribution:** 3
**Rating:** 4
**Confidence:** 4

**Summary:**

This paper proposes SWDG, an ensemble-based offline RL method that aims to improve uncertainty quantification robustness by reducing inter-critic correlation in Q-ensembles. The paper's empirical results show the proposed method achieves a state-of-the-art performance on D4RL mujoco and adroit tasks.

**Strengths:**

- The paper addresses an important problem in using an ensemble of deep neural networks, which is to reduce the correlation between each member of the ensemble.
- The paper provides a theoretical grounding of the proposed method.
- The paper shows strong empirical performance on the considered benchmarks.

**Weaknesses:**

- The writing needs to be improved to make the readers understand the method more clearly. Especially, a few of the used symbols are not defined before their usage (e.g., delta, u, and Ut).
- There could be many alternatives for introducing temporal asynchrony by updating only a subset of the Q-networks. For example, the simplest alternative is to uniformly randomly sample k Q-networks for each iteration. These kinds of alternative design approaches should be compared to justify the proposed approach.
- The paper argues that the previous methods (like EDAC) have limitations in breaking the correlation between the Q-networks. But the paper does not actually provide evidence on whether that argument holds empirically (or theoretically).
- As far as I know, the baselines CQL and IQL use the same hyperparameter setting for the D4RL mujoco tasks. However, Table 4 in supplementary shows the proposed method used different hyperparameter settings for each task. This is not a fair comparison.

**Questions:**

- Since the proposed method consists of two contributions (sliding-window temporal asynchrony and delayed-gradient targets), I would like to see an ablation study on when only one of the two contributions is applied. This would be helpful to grasp which of the two contributions is more critical.
- I would like to see experiment results on D4RL kitchen, which is also a challenging task like Adroit.

---

### Official Review · Reviewer_Xx5x · 2025-11-06

**Soundness:** 2
**Presentation:** 2
**Contribution:** 3
**Rating:** 4
**Confidence:** 3

**Summary:**

The paper proposes Sliding Window Delayed Gradient (SWDG): at each step, only a contiguous window of size m within an ensemble of N Q-critics is updated, while TD targets are built via a delayed subset (“delayed-min”) that is temporally offset from the currently updated critics. Within the window, the method also applies the EDAC cosine diversity loss. Figures and Algorithm 1 illustrate the pipeline; theory provides a pessimistic lower bound under boundedness and an upper bound on temporal decorrelation. Empirically, results are reported on D4RL Gym (MuJoCo) and D4RL Adroit.

**Strengths:**

1. Clear pipeline: sliding-window update + delayed-subset targets, with a figure and algorithmic description that make the method easy to implement.
2. Positive results on many MuJoCo tasks and several Adroit tasks; ablations show the delayed-gradient target helps on medium/medium-replay; runtime improves with smaller windows.

**Weaknesses:**

### 1) Novelty is thin

* The core move is a **sliding window** over critic ensembles plus a delayed-min target; within the window they also keep EDAC’s cosine diversity loss. That combination feels incremental relative to prior ensemble ideas (e.g., subset-min in REDQ, diversity in EDAC, independent targets in EQL).
* The paper **itself** notes the delayed-gradient effect is similar to using a **target network with update lag**, which undercuts the “newness” of that piece.
* Efficiency gains mainly come from computing **O(m) instead of O(N)** critics per step — an engineering choice rather than a new principle of uncertainty estimation.
### 2) Experiments are not enough
* **Benchmarks are narrow**: results are only on D4RL **MuJoCo** (HalfCheetah/Hopper/Walker2d, multiple dataset flavors) and **Adroit**; there’s no AntMaze/Kitchen or other harder/long-horizon suites in the presented tables.
* **Ablations are limited**: we see a window-size sweep (Hopper only) and a with/without delayed-gradient toggle; broader sweeps (e.g., step size *k*, ensemble size *N*, correlation sensitivity) across tasks are missing.

**Questions:**

Refer to Weaknesses.

---

### Note · Authors · 2025-11-12

I have read and agree with the venue's withdrawal policy on behalf of myself and my co-authors.